# Prevention of Post-Transplant Diabetes Mellitus: Towards a Personalized Approach

**DOI:** 10.3390/jpm12010116

**Published:** 2022-01-15

**Authors:** Didier Ducloux, Cécile Courivaud

**Affiliations:** 1CHU Besançon, Department of Nephrology, Dialysis and Renal Transplantation, Federation Hospitalo-Universitaire INCREASE, 25000 Besançon, France; ccourivaud@chu-besancon.fr; 2UMR RIGHT 1098, INSERM-EFS-UFC, 1 Bd Fleming, 25000 Besançon, France

**Keywords:** post-transplant diabetes, kidney transplantation, prevention

## Abstract

Post-transplant diabetes is a frequent complication after transplantation. Moreover, patients suffering from post-transplant diabetes have increased cardiovascular morbidity and reduced survival. Pathogenesis mainly involves beta-cell dysfunction in presence of insulin resistance. Both pre- and post-transplant risk factors are well-described, and some of them may be corrected or prevented. However, the frequency of post-transplant diabetes has not decreased in recent years. We realized a critical appraisal of preventive measures to reduce post-transplant diabetes.

## 1. Introduction

Post-transplant diabetes mellitus (PTD) is one of the most frequent complication safter kidney transplantation. However, frequency is still difficult to define. Before the international 2003 consensus [1], both incidence and prevalence of PTD were highly variable from one study to another. These discrepancies were in part inherent to patients’ heterogeneity and variations in immunosuppressive protocols, but were also due to the lack of uniform definition [2].

Nevertheless, the incidence of PTD remains highly variable in more recent studies using the international 2003 consensus definition of PTD (9 to 39%) [2,3]. Indeed, beyond differences in populations, diagnostic testing strategy also explains a large part of the heterogeneity between studies. Choice of method, timing, and frequency of metabolic controls highly influence the measured incidence of PTD. However, even when considering the lowest observed rate, at least one-in-ten KTR experiences PTD.

PTD is not only frequent, but is also associated with severe adverse outcomes and death. Thus, PTD is a risk factor for major atherosclerotic events [4] and most studies identified an association between PTD and premature death [5,6,7,8]. Association with graft failure is more controversial but observed in a number of cohort studies [8,9,10].

Thus, both frequency and severity of PTD justify a scientific questioning on PTD prevention. Nevertheless, the establishment of a prevention strategy first needs prerequisites including relevant identification of at-risk patients, solid knowledge of pathogenesis, and early detection tools. These points are crucial and need to be clarify before any prevention attempt. Second, prevention must be easy to manage, acceptable, well tolerated, and cost-effective.

## 2. Rationale for Prevention

### 2.1. Identification of at-Risk Patients

The first step of efficient prevention is appropriate selection of at-risk patients. Universal prevention cannot be an option. Both cost and unjustified exposition of low-risk patients to additional treatments imply a focused approach.

Many factors may contribute to PTD, but there are relatively few scores to classify patients. However, prediction models are essential tools for precision medicine. These models must accurately predict PTD to avoid giving unnecessary prevention in low-risk patients. Ideally, they must be available pretransplant to allow both pretransplant and immediate post-transplant prevention.

#### 2.1.1. Metabolic Evaluation

The clinical intuition mainly based on body mass index (BMI) or central obesity is not accurate. The predictive value of BMI or obesity is poor [11]. Measurement of visceral fat is a better risk indicator than BMI [12]. Moreover, a deep-learning-based quantification of visceral fat was found to be not only predictive of diabetes but above all enhanced the predictive value of a model based on classical risk factors [13]. Measurements of other key elements of the metabolic syndrome (triglycerides, HDL cholesterol, FPG) may be also more relevant than BMI.

The predictive value of abnormal pretransplant fasting glucose or an oral glucose tolerance test is important and should be systematically included in pretransplant screening. Pretransplant impaired fasting glucose (IFG) between 100 and 126 mg/dL and impaired glucose tolerance (IGT-2h between 140 and 200 mg/dL) strongly increase the risk of PTD [14,15,16,17,18]. Such patients with minor pretransplant abnormalities in glucose metabolism would benefit from targeted prevention.

While on the transplant list, patients should have yearly fasting plasma glucose and glycated haemoglobin tests. In those identified as at risk, annual oral glucose tolerance testing should be performed to detect pretransplant diabetes and impaired glucose tolerance.

#### 2.1.2. Genetics

Genetic susceptibility to PTD has been studied extensively [19]. Such studies have identified a large number of polymorphisms as being associated with PTD. Nevertheless, few studies met the criteria for SNP studies and even less were confirmed in replication studies [19]. Moreover, complementary analyses showing that being a genetic candidate adds to the predictive value of a risk model are even rarer. Finally, most genetic studies did not use the international 2003 consensus definition. 

We reported that IL-6 gene promoter polymorphism at position −174 was associated with PTD in two independent cohorts [20]. Although its predictive value in the whole KTR population is poor (20%), it reaches 50% in obese patients, allowing for a relevant clinical utilization. TCF7L2 rs7903146 also improves the predictive ability of a model including age, BMI, and previous transplantation [21]. A number of other studies have reported associations between different gene polymorphisms including CCL2, leptin, and angiotensinogen, and PTD [22,23,24].

Although of potential interest, the use of genetics parameters cannot be recommended. GWAS studies using a modern definition of PTD are an essential prerequisite to move in that direction.

By contrast, a familial history of diabetes is easy to collect and could be relevant for predicting the risk of PTD [25].

#### 2.1.3. Integrative Score

Some years ago, Chakkera et al. developed a simple and robust score to predict PTD [25]. Seven parameters including age, body mass index, familial history of diabetes, planned corticosteroid maintenance post-transplant, use of gout therapy, fasting glucose level, and fasting triglyceride concentration allow for discriminating three groups of patients. Half of those in the highest risk category developed PTD.

This score was secondarily validated in an independent cohort [26]. Nevertheless, it should be outlined that both characteristics of the studied population and choice of immunosuppression could mitigate the expected incidence estimated by this risk score. Thus, even when patients can be classified according to their own risk of diabetes, individual risk is difficult to accurately estimate. 

### 2.2. Pathogenesis

PTD is a clinical entity, which is distinct from type 1 and type 2 diabetes.

The pathogenesis is based on beta-cell dysfunction in the presence of insulin resistance. Consequently, prevention should try to protect beta-cell function and integrity and to enhance insulin sensitivity. Several data plead for a central role of beta-cell dysfunction as the predominant defect. First, GWAS identified different genes implicated in beta-cell apoptosis as being associated with PTD [19]. Second, during a prospective study, Hagen et al. reported that decline in insulin secretion was the principal determinant of PTD onset [27].

Although many factors participate in PTD pathophysiology, a simplified view is that immunosuppressive drugs determine the occurrence of PTD in a predisposed patient. CNI, and more particularly tacrolimus, play a major role in PTD [2]. Tacrolimus decreases insulin secretion with exacerbated consequences in patients with previous insulin resistance [28] (Figure 1). CNI induce upregulation of insulin gene expression and a decrease in insulin synthesis by transcriptional inhibition of insulin mRNA [29]. Such triggers in a predisposed patient explain a large part of PTD.

### 2.3. Diagnostic Tools

Many patients exhibit early postoperative hyperglycemia. Even when such patients are likely to be at risk of PTD, diagnosis can only be made after a period of 6 weeks post-transplant in a stable patient [30].

The oral glucose tolerance test is the gold standard for the diagnosis of PTD [30]. However, this test suffers from weak reproducibility and is not easy to organize.

Fasting blood glucose is easy to perform but is reported to miss about one-third of patients with IGT.

Glycated hemoglobin (HbA1c) is a diagnostic tool in nontransplant diabetes, but its utility in PTD diagnosis remains debated. HbA1c should not be used in the first three months post-transplant because of the frequency of clinical interferences (blood loss, blood transfusions, iron deficiency, acidosis, …). In a study including 199 patients transplanted for at least 3 months, Hoban et al. reported that among 20 patients with HbA1c > 6.1%, 14 had normal FPG [31]. This suggests that in stable KTR, HbA1c is more sensitive than FPG. In a more recent study including patients after 3 months post-transplant, HbA1c > 6.5% had a diagnostic concordance of 88.9% with abnormal OGTT [32]. Furthermore, HBA1c < 6.5% was associated with a normal OGTT in 98.7% [32].

Even when OGGT is the reference tool, HbA1c can be used in stable KTR after 3 months post-transplant. 

## 3. Prevention

The field of potential interventions is very large, ranging from the simplest to the most sophisticated. Prevention measures mainly include lifestyle modifications. The main issue concerning this strategy is to switch from a soft incentive to an active program for each at-risk patient. A popular strategy for transplant physicians is to adapt immunosuppression to reduce the risk of diabetes. However, this attitude is not without risks and most guidelines recommend avoiding such policy. Finally, although still a field of research, pharmacological prevention of PTD is an exciting and promising hope for tomorrow (Figure 2). 

### 3.1. Dietary Interventions

Based on risk factors for PTD and on experience in type 2 diabetes mellitus, lifestyle interventions are supposed to be the first-line approach for both prevention and treatment of PTD.

There are only a few pretransplant dietary interventions reported in the literature. Consequently, whether such a strategy is relevant in the primary prevention of PTD remains elusive. In fact, most studies concern post-transplant interventions.

Thus, the number of vegetables consumed was inversely associated with PTD in a prospective, longitudinal, observational study including 432 KTR [33]. Variations in components of metabolic syndrome mainly explained the effect of vegetable intake on PTD occurrence. 

Another study reported lower incidence of PTD in patients with a more pronounced Mediterranean-style diet [34].

Such a dietary approach is not easy to transpose to dialysis patients. The latter often have dietary constraints including water and potassium restriction that limits fruit and vegetable consumption. 

In fact, most pretransplant approaches focus on weight loss, specifically targeting obese patients. Weight reduction is controversial in dialysis patients. Although weight loss is frequently required for obese patients to access the waiting transplant list, many factors prevent the application of a dietary program. First, overweight dialysis patients have an apparent survival advantage that can discourage physicians [35,36]. It is difficult to decide whether this paradoxical association, also called “reverse epidemiology”, reflects a biological reality or is due to multiple biases of analysis.

Second, there is no clear survival advantage associated with weight loss in dialysis patients, even after transplantation [37]. Finally, results of a slimming diet are supposedly disappointing in dialysis patients. Nevertheless, a recent meta-analysis found that weight-loss interventions, mostly consisting of lifestyle modifications, are effective in reducing body weight and fat mass [37].

Some key elements are important to consider. First, BMI is a poor marker of metabolic risk. Better parameters including measurement of waist circumference of visceral fat, and recognition of metabolic syndrome, should advantageously replace or be added to BMI [12]. Second, physicians should be actively implied in dietary programs and the strategy must be both personalized and multidisciplinary [38]. It is probably not sufficient to encourage patients, or to give nonpersonalized leaflets and general recommendations. A recent study reported that only active lifestyle intervention provides weight loss and reduction in fat mass after kidney transplantation [39]. Additionally, PTD tends to be less frequent in patients having received active care [39]. 

### 3.2. Bariatric Surgery

There are several reports of bariatric surgery experience in CKD patients, including those on a kidney transplant waiting list [40,41,42]. Bariatric surgery enhances the access to transplant while improving pretransplant health status [40,43]. Postoperative mortality is very low [44]. Furthermore, sleeve gastrectomy decreases the prevalence of diabetes in CKD patients [40]. Kim et al. reported that PTD tended to be less frequent in 20 KTR having underwent pretransplant laparoscopic sleeve gastrectomy than in 40 controls matched for age, gender, and BMI [44].

Nevertheless, there are still few studies focusing on prevention of PTD using pretransplant bariatric surgery in obese candidates for kidney transplantation. Furthermore, the main interest of bariatric surgery is to allow access to transplantation to patients whose overweight is a contraindication to KT.

### 3.3. Physical Activity

CKD patients often have low levels of physical activity. A number of factors including socioeconomics conditions, sedentary life, comorbidities, and uremia-related asthenia participate to the weak physical activity. Although a large number of studies reported the beneficial effects of exercise in CKD patients, few data are available concerning the potential consequences on PTD [45,46,47].

Exercise enhances muscle glucose uptake through different mechanisms, some of them being insulin-dependent [48]. High-to-moderate physical activity stimulates the activation of 5-alpha-AMP-activated kinases (AMPKs). AMPKs’ activation mainly depends on the elevation of the ratio AMP/ATP during exercise. AMPKs increase glucose uptake and fatty acid oxidation. Low-intensity exercise also activates AMPKs but through intracytoplasmic calcium elevation. Exercise-induced glucose uptake is mainly dependent on GLUT4 translocation from intracellular storage sites to the cellular membrane. Enhanced muscle glucose uptake is a major component of PTD prevention by physical exercise.

Furthermore, exercise allows for weight reduction and induces changes in diet [49].

Furthermore, IL-6 is released by muscle during exercise. Contrary to most other sites of release, IL-6 produced by muscles depends on JUN N-terminal kinase and activator protein 1 signaling, and exhibits anti-inflammatory properties [50,51]. However, both exercise and IL-6 infusion suppress inflammation induced by endotoxin injection [52]. Modulation of cytokine secretion by muscles during exercise may support a link between physical activity, inflammation, and diabetes.

Different meta-analyses have reported that physical exercise associated with dietary intervention prevent overt diabetes in patients at risk [53,54,55]. However, the effects of physical exercise as a single intervention are less convincing [56,57]. Few encouraging post-transplant data are available. Moderate-to-vigorous physical activity after transplantation is associated with a lesser incidence of PTD [58]. Walking is a safe and easy to accept type of exercise that can be realized by most patients on the transplant waiting list. Such physical activity is sufficient to reduce HbA1c and body mass index in patients with type 2 diabetes [59]. At the opposite end, high-intensity exercise could be an alternative for eligible patients. These protocols are time-sparring and have been proven to reduce blood glucose in type 2 diabetes obese patients [60].

### 3.4. Microbiota

With the development of DNA-sequencing platforms, research on microbiota has emerged in several health domains, including diabetes [61]. Concordant results suggest that dysbiosis is associated with onset and progression of type 2 diabetes [59]. Reduced diversity and increase in Firmicutes/Bacteroidetes ratio are the most recognized specificities of diabetes-related microbiota [62].

Dysbiosis interacts with glucose metabolism through different mechanisms including variations in production of short-chain fatty acids and branch-chain amino acids, increase in endotoxemia-induced septic low-grade inflammation, alteration in the gut barrier, and dysregulation of the bile acid pool size and composition.

Even more interesting is the fact that patients with prediabetes exhibit changes in microbiota [63,64]. Butyrate seems to play a pivotal role in the interplay between gut microbiota and diabetes. A reduced abundance of butyrate-producing bacteria is associated with diabetes. Furthermore, metabolites derived from butyrate affect insulin sensitivity [65].

Microbiota manipulation including butyrate-enriched diet may be interesting in predefined patients carrying a high risk of PTD.

### 3.5. Pharmacological Approach

Beta-cell dysfunction occurring in a predisposed patient, is the main trigger of PTD. The use of drugs with protective effects on beta pancreatic cells is a fascinating research approach. Although numerous experimental data support the strong interest of such strategy, few clinical data are still available.

Two non-mutually exclusive options exist. One is to target high-risk patients before transplantation for primary prevention. The second is to intensively treat patients who develop early postoperative hyperglycemia. In the first case, the aim of treatment is to prevent CNI-induced beta-cell dysfunction. In the second, the objective is to reduce and/or prevent glucotoxicity of beta cells. However, the two strategies may be associated.

#### 3.5.1. Glucotoxicity and Beta-Cell Protection

Beta-cell failure may correspond to different situations which are not mutually exclusive [66]. The first one is due to a reduction in beta-cell mass. In humans, even transient repeated hyperglycemia induces beta-cell apoptosis [67]. In addition, patients with impaired glucose tolerance have reduced beta-cell mass before overt diabetes [68]. Beta-cell mass may also depend on genetic or environmental factors that may aggravate the consequences of hyperglycemia and precipitate the onset of diabetes [69]. The second one corresponds to beta-cell exhaustion. Hyperglycemia and/or hyperlipemia may induce endoplasmic reticulum stress that results in beta-cell exhaustion. Beta cells are histologically normal but cannot secrete insulin. This state is reversible with intensive metabolic control [70,71]. Furthermore, caloric restriction in newly diagnosed type 2 diabetes patients restores beta-cell function and normalizes glycemic control [72,73]. Beta cell dedifferentiation or transdifferentiation represents the last situation. Talchai showed that mice lacking FOX1 have hyperglycemia and reduced beta-cell mass. Reduction in beta cell mass was not due to cell death, but to beta-cell dedifferentiation [74].

Glucose toxicity on beta-cells is an unquestionable certainty. Therefore, early intensive treatment of hyperglycemia may prevent irreversible damage and favor self-metabolic control recovery. In newly diagnosed type 2 diabetes patients, a short course of continuous sc insulin infusion is able to allow good glycemic control with diet alone in about 50% of cases in one year [75,76]. Recovery of beta-cell function after intensive insulin treatment is predictive of future metabolic autonomy [76].

#### 3.5.2. Pharmacological Interventions to Prevent PTD

Hecking et al. first studied the effects of early insulin post-transplant therapy on long-term metabolic control autonomy [77]. Fifty KTR were randomized 1,1 to have either strict glycemic control with insulin (goal 110–120 mg/dL) or a more wait-and-see approach (thresholds for treatment > 180 mg/dL). One-year post-transplant, all the patients included in the intensive treatment group were free of antidiabetic treatment, whereas 28% of controls received such therapies. Beta-cell function was also better in patients having received insulin.

More recently, the same group performed an open study and compared the use of intermediate-acting insulin for postoperative afternoon glucose > 140 mg/dL to short-acting insulin for postoperative fasting glucose > 200 mg/dL [78]. In the intention-to-treat analysis, the incidence of PTD was similar in the two groups. Nevertheless, in per-protocol analysis and after adjustment for polycystic disease, PTD was less frequent in the intensive group two years post-transplant. However, more hypoglycemic episodes occurred in the treatment group. Although somewhat disappointing, this study suggests that the concept of beta-cell protection from glucose toxicity applies to PTD.

The lessons that can be drawn from this study must allow preventing some pitfalls. First, adherence to treatment is critical. In the latter study, nonadherence to insulin therapy was associated with a greater incidence of PTD. In that respect, the treatment should be easy to administer. Second, patients assigned to the intensive treatment group experienced more hypoglycemic episodes. The preventive treatment should be well-tolerated. 

Gliptins could be an interesting alternative to insulin. These drugs inhibit dipeptyl peptidase-4 (DPP-4), which inactivates the incretins, the glucagon-like peptide-1 (GLP-1), and the gastric inhibitory polypeptide (GIP). DPP-4 inhibition results in an increase in GLP-1 and GIP concentrations which induces insulin release and inhibition of glucagon secretion. 

GLP-1 has well-established proliferative and antiapoptotic effects on beta cells [79]. The effects of GLP-1 include expansion of beta-cell mass and resistance to beta-cell injury in experimental models of diabetes in vivo. Thus, GLP-1R agonists prevent or delay the development of diabetes in both db/db mice and Goto-Kakizaki rats, and diminish the severity of diabetes in rats after partial pancreatectomy or neonatal administration of streptozotocin [80,81,82,83]. However, DPP-4 is widely expressed on many immune cells, including monocytes, natural killer cells, and T lymphocytes, and is supposed to regulate their activation and chemotaxis [84]. Inhibition of DPP-4 reduces the release of different cytokines, including IL-2, IL-12, and interferon-γ. Furthermore, Transforming Growth Factor-β is upregulated after inhibition of DPP-4 [84].

Of interest, chronic low-inflammatory state is associated with the occurrence of PTD [20]. Finally, gliptins are well-tolerated and do not promote hypoglycemia.

Vildagliptin was used in the treatment of diabetes after kidney transplantation in two clinical studies [85,86]. Both studies indicate good safety and efficacy, suggesting that this drug may be widely used in KTR.

We recently began a double-blind, randomized, controlled study testing whether a short-term treatment by Vildagliptin in the early post-transplant period may prevent PTD [87]. We hope to achieve concrete results next year.

### 3.6. Individualization of Immunosuppression

As many immunosuppressive drugs are implicated in PTD occurrence, a number of studies have attempted to prove that changes in immunosuppressive protocols might decrease PTD incidence. Nevertheless, it appears from literature that it is difficult to decrease the incidence of PTD without increasing the incidence of acute rejection. Thus, most recommendations suggest that the choice of immunosuppression should be made to prevent acute rejection rather than to decrease PTD incidence. This firstly implies that changes in immunosuppression based on risk of diabetes must only concern patients at low immunological risk. Furthermore, any change made for PTD prevention should be balanced by another change to maintain equivalent immunosuppression. 

#### 3.6.1. Steroid-Sparing Protocols

The effects of steroid-sparing protocols on PTD incidence are the object of debate. A meta-analysis reported no effect of late cessation of steroid treatment (3–6 months post-transplant) on PTD occurrence, while acute rejection was found to be more frequent in patients receiving CsA [88]. A more recent meta-analysis by the Cochrane database concluded that steroid withdrawal or avoidance increased the risk of acute rejection with only a marginal reduction in PTD incidence [89]. Nevertheless, in the HARMONY study, rapid cessation of steroids was associated with a decreased incidence of PTD without any increase in the risk of acute rejection [90].

As steroids are undoubtedly diabetogenic, the discrepancies between studies are likely to be explained by different external factors. First, CNI increase the risk of PTD and high dosage of these drugs in a context of steroid-sparing may mask the benefits of this strategy. Second, early withdrawal of steroids after transplantation warrants greater effects on PTD but further increases the risk of acute rejection. Consequently, a global strategy including steroid withdrawal, CNI minimization, and induction with lymphocyte depletive therapy, offers the better-combined results.

#### 3.6.2. Cyclosporin vs. Tacrolimus

Even if all CNI are diabetogenic, many studies demonstrated that tacrolimus carries a greater risk of PTD than CsA [91]. Furthermore, both in vitro and in vivo, tacrolimus reduces insulin secretion to a greater extent than CsA [91].

However, even when CNI are the cornerstone of immunosuppression in KT, tacrolimus is more efficient to prevent acute rejection and is much more widely used than CsA [92]. Consequently, the problem is not so much CNI avoidance but substitution of tacrolimus by CsA. The use of CsA to reduce PTD risk can be part of two distinct strategies. The first one consists of de novo use in high-risk patients. The second one is based on conversion of tacrolimus for CsA in patients who have developed PTD.

A randomized study published 15 years ago compared CsA and tacrolimus for PTD occurrence [93]. PTD was less frequent in CsA-treated patients and the incidence of acute rejection was similar with the two drugs. However, in this already old study, doses of tacrolimus were higher than they are today, and the follow-up was short (6 months). Furthermore, external validation is poor since almost all studies comparing CsA and tacrolimus reported higher risk of acute rejection with CsA [89].

Thereafter, two meta-analyses have confirmed the increased diabetogenic effect of tacrolimus [94,95]. Nevertheless, acute rejection was more frequent in CsA-treated patients [95]. A recent study reported that while PTD was three-fold more frequent in tacrolimus-treated patients than in those receiving CsA, and transplant failures including death, graft loss, and acute rejection, were two-fold more frequent in CsA-treated patients [96]. The second strategy aims to use tacrolimus de novo and to consider conversion to CsA in patients who developed PTD. Optimal prevention of acute rejection is preserved, and prevention is even more targeted. However, whether tacrolimus-induced PTD is reversible after a switch towards CsA is questionable. 

A recent RCT carries a clear answer. Eighty KTR with PTD were randomized to either continue tacrolimus or receive CsA instead of tacrolimus. One-year after, 39% of patients in the CsA arm were free of diabetes vs. 10% in the TAC arm (*p* = 0.01) [97]. Even when the number was low and hampered definite conclusions, the incidence of acute rejection was similar in the two groups.

#### 3.6.3. mTOR Inhibitors

Numerous studies documented the effects of mTORi on glucose metabolism. mTORi interferes with insulin signaling through inhibition of the insulin receptor–IRS–PI3k–Akt pathway [98]. Furthermore, mTORi favors beta-cell apoptosis [99]. Teutonico et al. reported a decreased insulin sensitivity and a defect in beta-cell response after conversion from CNI to sirolimus [100]. Moreover, an analysis of the USRDS cohort including 20,154 first KT concluded that patients receiving sirolimus + CNI were at a higher risk of PTD [101]. In these former studies, sirolimus dosages were very high compared to current practices, and may have biased the results.

More recently, new schemes of utilization of mTORi were based on association with low doses of CNI. In this context, PTD incidence was similar in patients receiving either everolimus associated with low dose of TAC or MMF with standard dose of TAC [102].

Consequently, mTORi should not be considered to prevent PTD.

#### 3.6.4. Belatacept

Belatacept is a fusion protein composed of the Fc fragment of human IgG1 linked to the extracellular domain of cytotoxic T-lymphocyte-associated antigen 4 (CTLA-4), that selectively inhibits T-cell activation through costimulation blockade. Belatacept is the last approved immunosuppressive drug in organ transplantation. Recent data suggest that belatacept increases patient and graft survival in KTR with low-to-intermediate immunological risk [103].

In a pooled analysis of BENEFIT and BENEFIT-EXT studies, the incidence of PTD was reduced in belatacept-treated patients [104]. In a meta-analysis including 1049 patients with available data, the risk of PTD was reduced by 39% in patients treated by belatacept. The risk of acute rejection was only marginally increased. Of note, belatacept was mostly compared to CsA which would help explain the similar rate of acute rejection, but could minimize the beneficial effect on PTD occurrence [105].

Belatacept is an interesting option in patients at risk of PTD while being at low-to-intermediate risk of acute rejection.

#### 3.6.5. Global Immunosuppressive Strategies in High-Risk Patients

Very few immunosuppressive drugs are free of diabetogenic effects (Table 1). In patients with low immunological risk, belatacept-based therapy is a seducing option. In other patients, steroid sparing and CNI minimization with lymphocyte-depletive agents for induction is a safe strategy (Figure 3). It seems better to adapt the whole immunosuppressive regimen than to avoid a single immunosuppressant.

## 4. Conclusions

PTD is a frequent and serious complication after kidney transplantation. A strategy of prevention based on pathogenesis and patients’ phenotype is already available. Pretransplant lifestyle modifications and adaptation of immunosuppressive treatment when possible are at least theoretically easy to apply. These approaches must be at least discussed and put in place in high-risk patients. Intensive treatment of early post-transplant hyperglycemia is a reasonable recommendation regarding published data on type 2 diabetes and PTD. Pharmacological prevention of PTD is a subject of research with expected results in few years. There is an obvious need for all physicians involved in organ transplantation to address this important clinical issue. Progress in basic and clinical research in PTD prevention are still required.

## Figures and Tables

**Figure 1 jpm-12-00116-f001:**
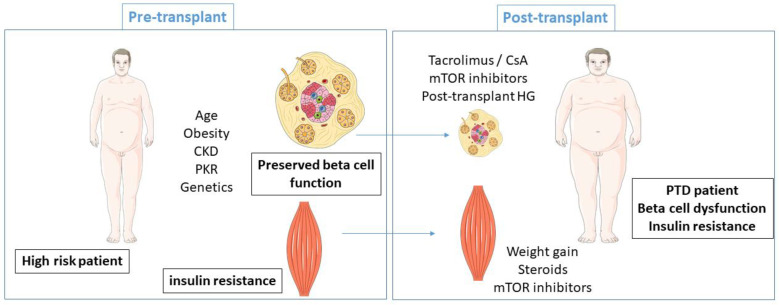
Summarized pathogenesis of post-transplant diabetes. (CKD, chronic kidney disease; HG, hyperglycemia).

**Figure 2 jpm-12-00116-f002:**
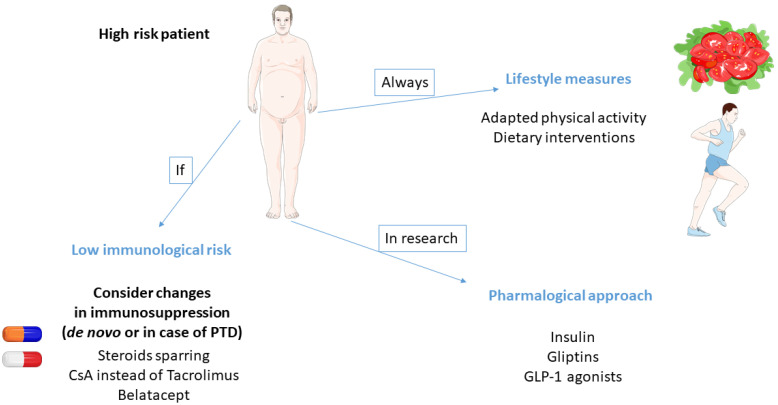
Summarized prevention strategies in post-transplant diabetes.

**Figure 3 jpm-12-00116-f003:**
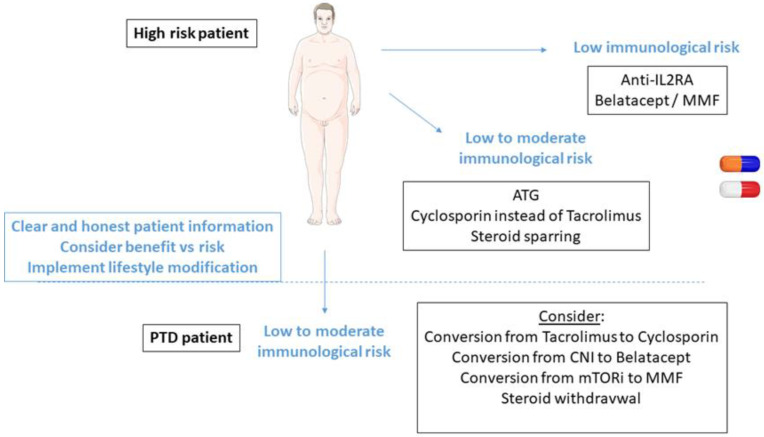
Strategies to prevent or control PTD by modulating immunosuppressive regimen; (ATG, antithymocytes globulins; CNI, calcineurin inhibitors; MMF, Mycophenolate Mofetil; mTORi, mTOR inhibitors; PTD, post-transplant diabetes).

**Table 1 jpm-12-00116-t001:** Effects of immunosuppressive drugs on glucose metabolism.

	Risk of PTD
Steroids	+++
Cyclosporin	++
Tacrolimus	+++
mTORi	++
AZA/MMF	0
Belatacept	0

AZA: Azathioprine/MMF: Mycophenolate Mofetil).

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
