# Peer review of "Prevention of Post-Transplant Diabetes Mellitus: Towards a Personalized Approach"

_jpm, 2022, doi:10.3390/jpm12010116_

Round 1

Reviewer 1 Report

The pathogenesis of insulin resistance is complex and not fully understood, especially in the case of development and Post-Transplant Diabetes Mellitus, therefore the publication presented for review is very interesting and important. The authors showed that the genetic factors overlap with the influence of the external environment, especially the administration of immunosuppressive drugs such as tacrolimus, cyclosporine and steroids, also increased energy supply with food, mainly derived from animal fats and easily digestible carbohydrates, and low physical activity.

The genetic factors associated with the development of Post-Transplant Diabetes Mellitus have been insufficiently described. The authors only mentioned the IL6 gene promoter polymorphism at position -174 AND TCF7L2 rs7903146. However, there are more studies on the relationship between genetic factors and the development of Post-Transplant Diabetes Mellitus. An example can be the analysis of CCL2 gene polymorphism, which is associated with post-transplant diabetes mellitus [Dabrowska-Zamojcin E, Romanowski M, Dziedziako V, Maciejewska-Karlowska A, Sawczuk M, Safranow K, Domanski L, Pawlik A. CCL2 gene polymorphism is associated with post-transplant diabetes mellitus. Int Immunopharmacol. 2016 Mar; 32: 62-65. doi: 10.1016 / j.intimp.2016.01.011. Epub 2016 Jan 20. PMID: 26802601.] Other studies show an association between the leptin rs2167270 gene A allele and PTDM development [Romanowski M, Dziedziako V, Maciejewska-Karlowska A, Sawczuk M, Safranow K, Domanski L, Pawlik A. Adiponectin and leptin gene polymorphisms in patients with post-transplant diabetes mellitus. Pharmacogenomics. 2015; 16 (11): 1243-51. doi: 10.2217 / pgs.15.71. Epub 2015 Aug 18. PMID: 26282401.], AGT gene rs4762 polymorphisms for the development of PTDM [Lee S, R, Moon J, Y, Lee S, H, Ihm C, G, Lee T, W, Kim S, K , Chung J, -H, Kang S, W, Kim T, H, Park S, J, Kim Y, H, Jeong K, H: Angiotensinogen Polymorphisms and Post-Transplantation Diabetes Mellitus in Korean Renal Transplant Subjects. Kidney Blood Press Res 2013; 37: 95-102. doi: 10.1159 / 000343404], the association of calpain-10 gene polymorphism and posttransplant diabetes mellitus in kidney transplant patients medicated with tacrolimus [Kurzawski, M., Dziewanowski, K., Kedzierska, K. et al. Association of calpain-10 gene polymorphism and posttransplant diabetes mellitus in kidney transplant patients medicated with tacrolimus. Pharmacogenomics J 10, 120-125 (2010). https://doi.org/10.1038/tpj.2009.44]. I propose to make a table with a compilation and significance of genetic factors.
In References, please delete the additional numbering that appeared from 17-112.

Author Response

Dear reviewer,

Thank you four your interesting remarks.

We added some datas and references in the paragraph "genetics of PTD".

Our references did not appear in the initial submitted form as they appear in the reviewed form. We hope that they are well numbered in the revised form

Reviewer 2 Report

It is recommended to summarize strategy for prevention of PTD in the figure.

Please unify to IL-6. Line72

The content is very interesting, but the structure of the paper seems to be unbalanced as follows;

The content of “2.1 Identification of at-Risk Patients is too long compared to 2.2 and 2.3.

In addition, some of the contents are duplicated and some items should be summarized. (For example, 3.5 and 3.6.2)

For immunosuppressive therapy in 3.6, please provide a subtitle that concisely describes the effect of these drugs on the development of PTD and add a discussion of how future immunosuppressive therapy should be planned based on the above results.

Author Response

Dear reviewer,

Thank you for your interesting comments.

We acknowledge that the content of the paragraph 2.1 is much longer than the next two. Nevertheless, there are so much to say in the first paragraph that we think that this difference is necessary.

I don't understand what is duplicated in 3.5 (pharmacological approach) and 3.6.2 (cyclosporin vs tacrolimus)

We added a paragraph, a table, and a figure to summarize the effects of immunosuppressive drugs
